# A Computational Magnetohydrodynamic Modelling Study on Plasma Arc Behaviour in Gasification Applications

**Quinn G. Reynolds [1,2,]*, Thokozile P. Kekana [1] and Buhle S. Xakalashe [1]**

1 Mintek, Private Bag X3015, Randburg 2125, South Africa
2 Department of Chemical Engineering, University of Stellenbosch, Private Bag X1, Matieland 7602, South Africa
* Correspondence: quinnr@mintek.co.za

**Abstract:** The application of direct-current plasma arc furnace technology to the problem of coal gasification is investigated using computational multiphysics models of the plasma arc inside such units. An integrated modelling workflow for the study of DC plasma arc discharges in synthesis gas atmospheres is presented. The thermodynamic and transport properties of the plasma are estimated using statistical mechanics calculations and are shown to have highly non-linear dependencies on the gas composition and temperature. A computational magnetohydrodynamic solver for electromagnetically coupled flows is developed and implemented in the OpenFOAM® framework, and the behaviour of three-dimensional transient simulations of arc formation and dynamics is studied in response to different plasma gas compositions and furnace operating conditions. To demonstrate the utility of the methods presented, practical engineering results are obtained from an ensemble of simulation results for a pilot-scale furnace design. These include the stability of the arc under different operating conditions and the dependence of voltage–current relationships on the arc length, which are relevant in understanding the industrial operability of plasma arc furnaces used for waste coal gasification.

**Keywords:** multiphysics; plasma; gasification

## 1. Introduction

The adaptation of direct-current (DC) plasma arc furnace technology from the metallurgical industry for use in the valorisation of low-grade waste coal material is currently in development at Mintek in South Africa. Discarded coal from mining operations is currently stored in large surface mine dumps in South Africa and elsewhere and represents a significant environmental and health hazard via effects, such as mine-drainage contamination of the local ground water and soil. A means of processing large quantities of such wastes together with contaminated water into valuable intermediate energy storage products, such as synthesis gas (syngas, a mixture of carbon monoxide and hydrogen) is, therefore, of some interest as this can potentially make the remediation of waste coal dumps more economically viable.

Such processes may also be combined with carbon capture and sequestration technologies to produce hydrogen for fuel and electricity-generation applications. The advantages of using DC furnaces in the gasification process include scaling to much larger units than traditional plasma torch gasifiers and the ability to treat waste carbon sources. Waste carbon includes discarded coal containing large quantities of ash and other impurities and ultra-fine coal fractions that are difficult to process in conventional applications. The concept of DC plasma furnace gasification is shown visually in Figure 1 and is described in more detail elsewhere [1].

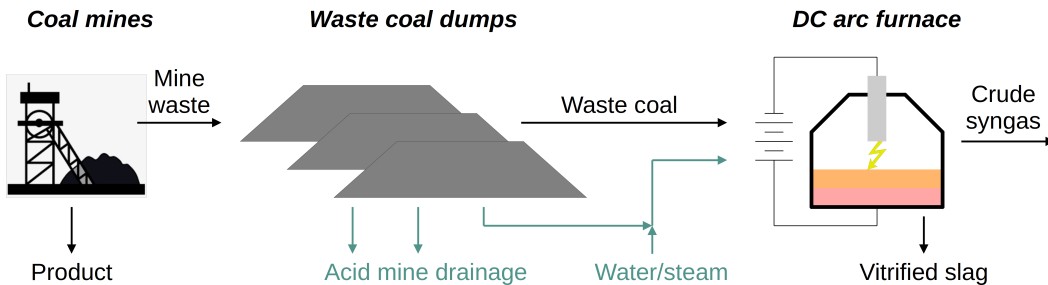

**Figure 1.** Application of DC arc furnace technology to the valorisation of waste coal using plasma gasification.

An important element in improving the engineering of DC plasma arc furnaces for applications, such as gasification, is a deep understanding of the behaviour of the plasma arc itself. Arcs are high-temperature, high-velocity jets of ionised gas that are generated and sustained by the close coupling between fluid flow, energy and electromagnetic fields when large electric currents are passed through thermal plasmas [2,3]. Due to the extremely large driving forces and numerous instability modes, arcs may exhibit a broad range of dynamics from steady state through to oscillatory and chaotic behaviour depending on the operating parameters [3–6].

As temperatures in the arc jet can reach well in excess of 10,000 K, direct experimentation on plasma arcs is extremely challenging. Mathematical models of various types are of value as proxies to help build intuition and understanding of arc behaviour under different operating conditions. The modelling of arcs can be divided into two main tasks—estimation of the thermodynamic and transport properties of the plasma in use and the application of empirical or computational models based on the fundamental governing equations of magnetohydrodynamic (MHD) systems to calculate the arc behaviour.

Plasma properties are generally estimated using detailed statistical mechanics calculations and can have highly non-linear dependencies on temperature and pressure for even the simplest noble gases [2]. The complexity increases further when multiple gases are present, such as the CO and $H_2$ mixtures typical of syngas. An example of the plasma dissociation and ionisation reactions to be considered is shown below for the hydrogen species only.

$$H_2 \rightarrow H + H \tag{1}$$

$$H_2 \rightarrow H_2^+ + e^- \tag{2}$$

$$H \rightarrow H^+ + e^- \tag{3}$$

In the full syngas mixture, the following species and the reactions between them are included: $CO$, $H_2$, $CH$, $OH$, $CO^+$, $H_2^+$, $CH^+$, $OH^+$, $C$, $C^+$, $C^{2+}$, $H$, $H^+$, $O$, $O^+$, $O^{2+}$ and $e^-$. This system will be examined further as part of the present paper.

Computational modelling of arc behaviour by numerical solutions of the differential equations of fluid flow, heat transfer and electromagnetism is a challenging problem and has advanced in lockstep with increasing computer power and research into numerical methods. The earliest efforts used simplified geometries and pre-calculated electromagnetic fields in otherwise standard computational fluid dynamics (CFD) models to produce steady state solutions, such as those presented by Szekely and colleagues [7].

Soon after this, fully coupled MHD-CFD models became possible and then transient solutions, extended physics and other effects. These efforts have culminated in computational multiphysics models capable of simulating the transient dynamics of three-dimensional arcs under a wide variety of conditions [8–10]. Despite this progress, a number of important effects are often neglected in arc models and remain active areas of research and development today.

Among these are the formulation of accurate and mesh-independent boundary conditions able to represent non-equilibrium plasma sheath effects and the ablation of electrode

and work material surfaces [11,12]. In addition, coupling between the electrode and plasma domains [13] and between the plasma and free-surface flows in the case of molten work materials [14] are significant in many cases but remain computationally expensive and algorithmically complex to implement. The convergence and computational cost of electromagnetic and thermal radiation solvers in coupled MHD-CFD models also remains a limiting step for performing high-fidelity transient simulations.

Despite the limitations on the state of the art, it was considered to be of interest to use modern plasma arc modelling tools to perform a preliminary examination of the behaviour of arcs in syngas mixtures. It is hoped that the methods presented during the course of this study may be of use for future work on arcs operating in unusual gas environments.

## 2. Model Description

As described above, the development of a practical model for plasma arcs operating in syngas mixtures requires two steps—calculation of the thermophysical properties of the plasma and numerical solutions of the governing equations.

### 2.1. Calculation of Plasma Properties

The basis for calculation of the material properties is a statistical mechanics description of the plasma and the assumption of a certain degree of thermodynamic equilibrium. Local thermodynamic equilibrium (LTE), in which it is assumed that particle collision processes happen very fast and a single temperature may be used to represent both heavy and light particles (molecules, atoms, ions and electrons), is frequently assumed. This is generally accepted to be a reasonable assumption in the case of atmospheric-pressure thermal plasma systems, such as plasma arcs [2], with the possible exception of the coldest outlying regions and the plasma sheaths near to the boundaries.

Under the assumption of LTE, the fundamental quantum mechanical properties of the particles present, such as their vibrational, rotational and electronic energy levels, determine their statistical partition functions, and the partition functions, in turn, define the thermodynamic quantities of a mixture, such as the chemical potential and Gibbs free energy.

Minimisation of the Gibbs free energy with respect to the particle concentrations using appropriate mass and charge balance constraints then yields an equilibrium plasma composition, from which thermodynamic material properties, such as the heat capacity $C_P$ and density $\rho$, may be determined [2]. For the present work, these calculations were performed using a module written for the Python programming language [15] combined with species data from the NIST Atomic Spectra and Chemistry WebBook databases [16,17].

In order to calculate transport properties, such as the viscosity $\mu$, thermal conductivity $\kappa$ and electrical conductivity $\sigma$, additional information about the particle collision processes in the plasma mixture is required. In particular, expressions for the collision cross sections and resulting collision integrals must be obtained for each particle pair and collision type. Once the collision integrals are known, Chapman–Enskog collision theory may be applied to determine the transport of various properties of interest. The procedure used in this work follows that presented by Devoto [18]. Generalised empirical and theoretical collision integrals were obtained from a variety of sources [19–21] and implemented for the various species present in $CO$-$H_2$ plasma mixtures.

### 2.2. MHD-CFD Multiphysics Model

The governing equations of a plasma arc system include the compressible Navier–Stokes and continuity equations for velocity **u** and pressure $P$, the energy transport equation for enthalpy $h$ (related monotonically to plasma temperature $T$) and Maxwell's equations for electric potential $\phi$ and magnetic vector potential **A**.

$$\frac{\partial(\rho\mathbf{u})}{\partial t} + \nabla \cdot (\rho\mathbf{u} \otimes \mathbf{u}) + \nabla P = \nabla \cdot \bar{\bar{\tau}} + \mathbf{j} \times \mathbf{B} - \rho\mathbf{g} \tag{4}$$

$$\frac{\partial \rho}{\partial t} + \nabla \cdot (\rho \mathbf{u}) = 0 \tag{5}$$

$$\frac{\partial (\rho h)}{\partial t} + \nabla \cdot (\rho \mathbf{u} h) = \nabla \cdot \left( \frac{\kappa}{C_P} \nabla h \right) + \nabla \cdot \left( \frac{5 k_B h \mathbf{j}}{2 e C_P} \right) + \frac{\mathbf{j} \cdot \mathbf{j}}{\sigma} + Q_m - Q_r, \tag{6}$$

$$\nabla \cdot \mathbf{j} = 0, \quad \mathbf{j} := -\sigma \left( \nabla \phi + \frac{\partial \mathbf{A}}{\partial t} - \mathbf{u} \times \mathbf{B} \right) \tag{7}$$

$$\nabla^2 \mathbf{A} = -\mu_0 \mathbf{j}, \quad \mathbf{B} := \nabla \times \mathbf{A} \tag{8}$$

In (4), $\bar{\bar{\tau}}$ is the viscous stress tensor, $\mathbf{j}$ is the current density vector (here representing the direction of flow of electrons, not positive charge carriers), and $\mathbf{B}$ is the magnetic field. In (6), $k_B$ is the Boltzmann constant, and $e$ is the fundamental charge. $Q_m$ is the mechanical source term due to heating by pressure and kinetic energy changes in compressible flows (in OpenFOAM's formulation, this is $\partial (\rho K) / \partial t + \nabla \cdot (\rho U K)$, where $K$ is the kinetic energy of the flow field). $Q_r$ is the thermal radiation source term obtained by solution of the radiative transport Equation (the formulation for this varies depending on the radiation model in use; however, for example, in the $P_1$ model, it is equal to $\alpha G - 4 \epsilon \sigma_{SB} T^4$, where $\alpha$ and $\epsilon$ are the material absorptivity and emissivity, respectively, $G$ is the radiation intensity field, and $\sigma_{SB}$ is the Stefan–Boltzmann constant). In (8), $\mu_0$ is the magnetic permeability of the plasma medium, taken as equal to the vacuum permeability.

Coupling exists in (4)–(8) due to the presence of the Lorentz momentum source term $\mathbf{j} \times \mathbf{B}$, the Ohmic heating term $\mathbf{j} \cdot \mathbf{j}/\sigma$, the induced current term $\mathbf{u} \times \mathbf{B}$ and the strong dependence of all material properties on temperature. In the present model, the only magnetic fields present are those that are self-induced by the current flow in the arc, although external fields could easily be applied in future studies. The action of the arc current, the induced magnetic field and the resulting Lorentz force term are shown in Figure 2.

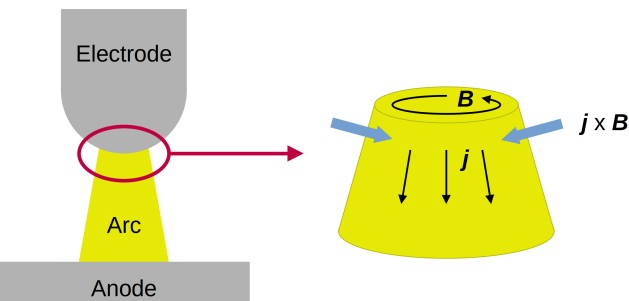

**Figure 2.** Schematic diagram of a DC plasma arc system, showing orientation of the $\mathbf{j}$, $\mathbf{B}$ and $\mathbf{j} \times \mathbf{B}$ vector fields.

The solution of the coupled set of MHD-CFD field equations is performed using the finite volume method in which the region to be modelled is decomposed into a mesh of many small control volumes, each of which enforces a problem-specific set of conservation laws with respect to its immediate neighbours. In the present study, the OpenFOAM® open source computational mechanics framework [22] was chosen to implement a solver for the plasma arc model.

The solver uses a segregated solution of the $\mathbf{u}$, $P$ and $h$ fields with a combined pressure-implicit splitting of operators (PISO) and semi-implicit method for pressure-linked equations (SIMPLE) algorithm to resolve the pressure–velocity coupling. An iterative segregated algorithm was implemented for the solution of the $\mathbf{A}$ and $\phi$ fields, combined with fast lookup tables for the material properties. Full details of the solver algorithm and implementation are available in previous work [23]. Adaptive time stepping with a Courant number limit of 0.95 was used for all simulations unless otherwise stated.

## 3. Results and Discussion

Calculations of the thermophysical properties of syngas plasmas are presented, followed by the results from MHD-CFD simulations of a small pilot-scale DC gasification furnace.

### 3.1. Material Properties for Syngas Plasmas

The gasification of a carbon source with water produces a mixture consisting primarily of CO and $H_2$ with some $CO_2$ and $CH_4$ present as contaminants. "Ideal" syngas contains only carbon monoxide and hydrogen in varying ratios. A range of 50% to 100% CO was chosen for the present study; pure CO is of interest as a reference as it is the typical gas environment in metallurgical reductive smelting processes that DC plasma arc furnaces are often used for.

At elevated temperatures, molecules become less theromdynamically stable compared with their constituent atoms, ions and electrons. A $CO/H_2$ plasma at moderate temperatures can, therefore, be expected to contain multiple intermediate species and decomposition products. As mentioned earlier, in these calculations, the following were included: CO, $H_2$, CH, OH, $CO^+$, $H_2^+$, $CH^+$, $OH^+$, C, $C^+$, $C^{2+}$, H, $H^+$, O, $O^+$, $O^{2+}$ and $e^-$.

As the system is assumed to be at LTE, no transport or chemical reaction kinetics are considered here. At a specified temperature, pressure and initial composition, the equilibrium mole fractions of each species (and, hence, the thermodynamic and transport properties) were calculated using the statistical mechanics methods described earlier. The results obtained for various syngas mixtures at one atmosphere pressure are shown in Figure 3.

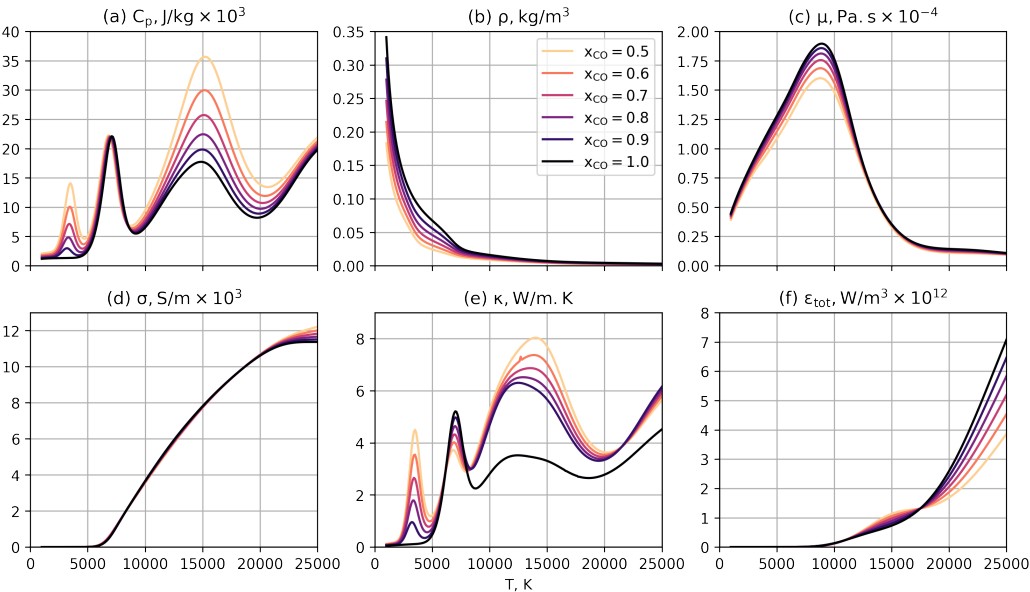

**Figure 3.** Material properties for syngas plasmas at atmospheric pressure: (**a**) Heat capacity. (**b**) Density. (**c**) Viscosity. (**d**) Electrical conductivity. (**e**) Thermal conductivity. (**f**) Volumetric total radiation emission coefficient.

Several effects are immediately obvious and have important implications for the behaviour of arcs in syngas environments. The first is that the properties are all very strong functions of temperature, which can be expected to reinforce the coupling between the thermal energy and other fields in the MHD-CFD model. Many of the properties also vary non-linearly and non-monotonically with temperature, especially $C_P$ and $\kappa$. The degree of non-linearity increases with the increasing hydrogen content in the mixture with a high peak developing at temperatures below 5000 K; this can be expected to introduce some additional sources of instability in the cooler outlying areas of the arc.

It is interesting to observe that the changing gas composition has relatively little effect on $\sigma$ except at extremely high temperatures. This suggests that the electrical behaviour of the arc would be similar over a range of syngas mixtures for a given temperature field (although, given the large differences in the thermal properties, it is likely that the temperatures will vary widely).

The total radiation emission curves show higher volumetric emission from $H_2$-rich syngas plasmas at intermediate temperatures, but this trend reverses at very high temperatures. Thermal excursions in the arc may, therefore, be somewhat less likely to be brought under control by negative feedback from the radiation source term in cases where more hydrogen is present.

*3.2. MHD-CFD Simulations*

In order to assess the impact of plasma properties on the behaviour of arcs in syngas mixtures, a MHD-CFD model of one of Mintek's DC furnace pilot plants was developed. This particular plant will be used for future testing of the DC furnace gasification concept and is supplied by a transformer and rectifier rated at 100 kVA nominal power (in practice, this delivers a maximum usable power of between 40 and 50 kW). The furnace unit consists of a graphite-lined cylindrical crucible connected to a bottom anode with a single graphite cathode electrode mounted vertically through the refractory-lined roof. The electrode is attached to a hydraulic mount and can be moved vertically. For context, some photographs of the pilot facility are shown in Figure 4.

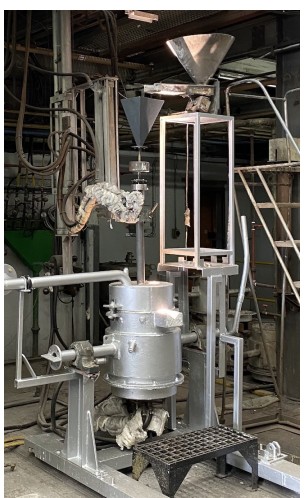 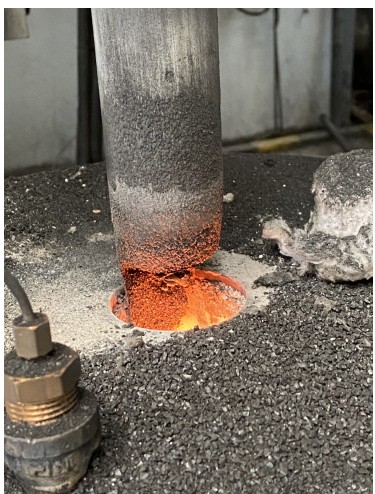

**Figure 4.** The 100 kVA DC plasma arc furnace pilot facility at Mintek. The furnace vessel is the silver cylinder at lower middle in the left image. A close-up of the electrode extracted from the furnace is shown on the right.

The dimensions, conditions and operating parameters used in the MHD-CFD model are given in Table 1. Unless otherwise specified, all models were initialised with a constant temperature of 10,000 K and a velocity field of zero and were ran for a total of 200 ms simulated time. For each simulation at a given arc length $L_a$ (defined as the clearance between the electrode tip and the anode surface below) and syngas CO mole fraction $x_{CO}$, the DC current was stepped down from 1000 to 200 A and back up again in steps of 200 A every 20 ms.

**Table 1.** Parameters for the 100 kVA furnace model.

| Parameter | Value | Parameter | Value |
|:---:|:---:|:---:|:---:|
| Region diameter | 0.2 m | Region height | 0.2 m |
| Electrode diameter | 0.05 m | Arc length $L_a$ | 0.01–0.05 m |
| DC current $I$ | 200–1000 A | CO fraction $x_{CO}$ | 0.5–1.0 |

Three-dimensional computational meshes consisting primarily of hexahedral elements were constructed for each arc length investigated. The meshes were successively refined in the central region between the electrode tip and the anode surface, where the arc jet is usually located. An example model geometry and mesh for the 0.05 m arc length case is shown in Figure 5.

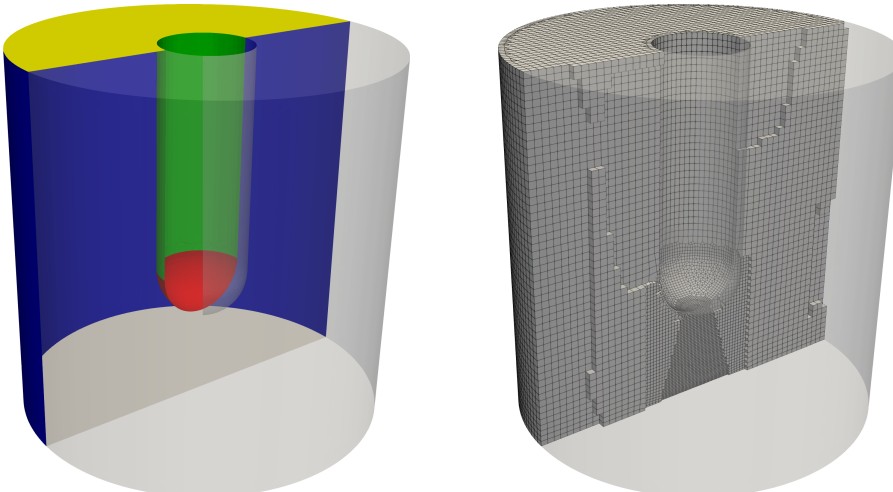

**Figure 5.** Example of model geometry and mesh used in MHD-CFD simulations. The boundary surfaces are shown at left—red is the cathode, grey is the anode, green is the electrode wall, blue is the vessel wall, and yellow is the vessel atmosphere.

Boundary conditions for the various fields in the model are given in Table 2, where **n** and **t** are normal and tangent vectors at the boundary surface, $j_k$ is the cathode spot current density (taken as 2 kA/cm$^2$ in this work), and $T_{lim}$ is a numerical limit placed on the surface temperature (typically the melting or vaporisation temperature of the boundary material). The boundary conditions for **A** represent a magnetically-insulating situation where the modelled region is surrounded by a highly conductive material, such as metal or graphite. The gradient boundary conditions for $\phi$ describe the current density on those surfaces, while the fixed value of zero represents the electrical ground or earth potential.

The conditional boundary conditions for $\phi$ and $h$ are described in [23] and are switched according to which portion of the cathode surface is acting as the arc attachment spot at any given time. This, in turn, is determined dynamically from the local temperature field in the plasma at each time step in the simulation.

**Table 2.** Boundary conditions for MHD-CFD model (see Figure 5).

| Field | Cathode | Anode | Walls | Atmosphere |
|:---:|:---:|:---:|:---:|:---:|
| **A** | $\frac{\partial(\mathbf{A}\cdot\mathbf{n})}{\partial\mathbf{n}} = 0$ $\mathbf{A}\cdot\mathbf{t} = 0$ | $\frac{\partial(\mathbf{A}\cdot\mathbf{n})}{\partial\mathbf{n}} = 0$ $\mathbf{A}\cdot\mathbf{t} = 0$ | $\frac{\partial(\mathbf{A}\cdot\mathbf{n})}{\partial\mathbf{n}} = 0$ $\mathbf{A}\cdot\mathbf{t} = 0$ | $\frac{\partial(\mathbf{A}\cdot\mathbf{n})}{\partial\mathbf{n}} = 0$ $\mathbf{A}\cdot\mathbf{t} = 0$ |
| $\phi$ | $-\sigma\frac{\partial\phi}{\partial\mathbf{n}} =$ $j_k/\frac{\partial\phi}{\partial\mathbf{n}} = 0$ | $\phi = 0$ | $\frac{\partial\phi}{\partial\mathbf{n}} = 0$ | $\frac{\partial\phi}{\partial\mathbf{n}} = 0$ |
| **u** | $\mathbf{u} = 0$ | $\mathbf{u} = 0$ | $\mathbf{u} = 0$ | $\frac{\partial\mathbf{u}}{\partial\mathbf{n}} = 0$ |
| $h$ | $T = T_{lim}/\frac{\partial T}{\partial\mathbf{n}} =$ $0$ | $T = T_{lim}/\frac{\partial T}{\partial\mathbf{n}} =$ $0$ | $T = T_{lim}/\frac{\partial T}{\partial\mathbf{n}} =$ $0$ | $\frac{\partial T}{\partial\mathbf{n}} = 0$ |

A limited mesh-dependence study was conducted to evaluate the MHD-CFD model's sensitivity to mesh resolution. For these tests, the simulations were run for only 10 ms at a fixed current and arc length of 1000 A and 0.05 m, respectively, in order to compare the initial transient behaviour. The arc voltage as a function of time was measured as the maximum of the $\phi$ field. The results are shown in Figure 6.

The finest two mesh resolutions behave quite similarly during the early phase of the simulations and capture the transition to transient dynamics shortly before 1 ms well. The dynamics of the finest three resolutions are similar during the latter stages of the simulations and show the system settling toward irregular oscillatory behaviour, whereas the coarsest resolution produces lower and steadier voltages. This is borne out by the quantitative results in Table 3, which show that, while the models are still not entirely mesh-independent, the finest three resolutions are well within one standard deviation interval of each other, and the results may, therefore, be considered as mesh-insensitive if not mesh-independent.

It is important to note that the MHD arc system is a chaotic transient flow, and only statistical comparisons are of value here—the exact trajectory of each case is likely to diverge exponentially after the initial conditions have decayed. The 1 mm resolution was chosen for the remaining simulations as a reasonable compromise of model performance and accuracy.

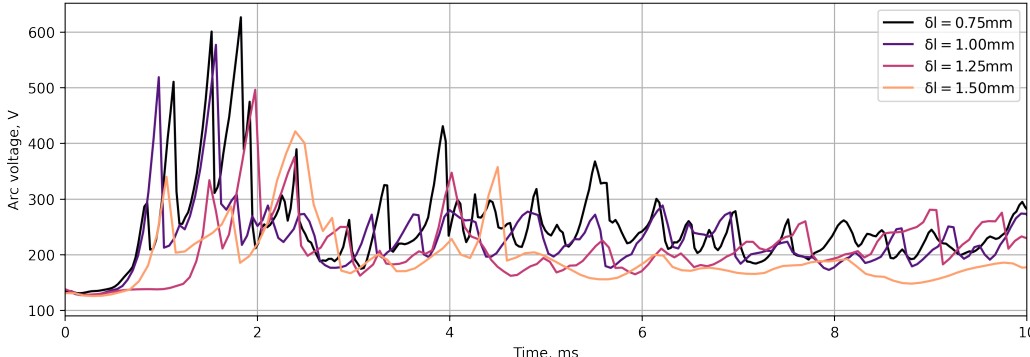

**Figure 6.** Initial evolution of arc voltages as a function of the minimum resolution in the mesh. $x_{CO} = 0.5$, $I = 1000$ A, $L_a = 0.05$ m.

**Table 3.** Arc voltages from mesh-dependence tests during the period of 5 to 10 ms.

| Resolution | Voltage Average | Voltage Std Dev. |
|---|---|---|
| 0.75 mm | 233.2 V | 34.0 V |
| 1.00 mm | 216.8 V | 28.6 V |
| 1.25 mm | 211.5 V | 29.1 V |
| 1.50 mm | 172.6 V | 12.8 V |

Although a rigorous model validation exercise must await data from future experiments on waste coal gasification using the 100 kVA test furnace or other facilities, a preliminary comparison was conducted against electrical models of DC plasma arcs originally developed by Bowman [3]. From observations of arcs in the 1–10 kA range, Bowman proposed a scale-invariant shape for the conducting core of the arc, which includes constriction effects near to the cathode attachment. By assuming a representative average plasma conductivity $\sigma_{avg}$ over the conducting core, the empirical shape function may then be integrated to obtain a relationship between the electrical variables (9).

$$V_a = \frac{1}{\sigma_{avg}} f(I, L_a) \tag{9}$$

Here, $f$, the integrated cell constant of the empirical arc shape, consists of several exponential decay terms and a linear term that depends only on $L_a$ [24].

It is important to note that the Bowman model applies only to steady, vertically-oriented arc jets. If any instability in the arc or asymmetry in its direction is present, the results of the two models cannot be easily compared. Validation was, therefore, performed using the parameters most likely to satisfy these conditions—the shortest arc length,

0.01 m, and the most stable plasma gas mixture, pure CO. The results of the analysis, using the average voltage in each current period, are shown in Figure 7.

The qualitative agreement is good, and the non-linear shape of the Bowman curve is captured well by the MHD-CFD model. Further work on validation against experimental data is, however, strongly advised to confirm that the computational model is working as expected.

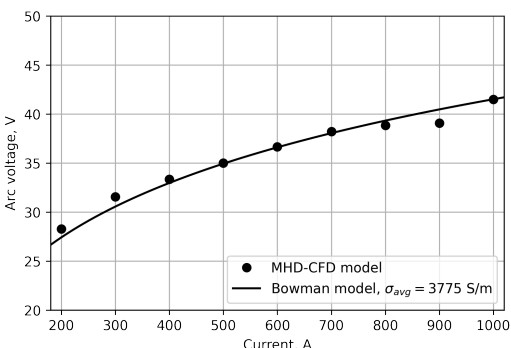

**Figure 7.** Comparison of computational model results with the Bowman empirical model. $x_{CO}$ = 1.0, $L_a$ = 0.01 m.

In order to explore the parameter space of the problem, a set of 30 simulations was then performed covering a range of different arc lengths and syngas compositions as indicated in Table 1. As described earlier, the simulations also varied the current to the arc in discrete steps every 20 ms. Two such current sweeps were performed during each simulation—decreasing from 1000 to 200 A and then increasing back up to 1000 A. Quantitative data in the form of the arc voltage were sampled from the model during each run. An example of this is shown in Figure 8. It is interesting to observe the noticeable changes in the arc dynamics during each current interval, in particular the transition from regular oscillations to a chaotic regime as the current increases.

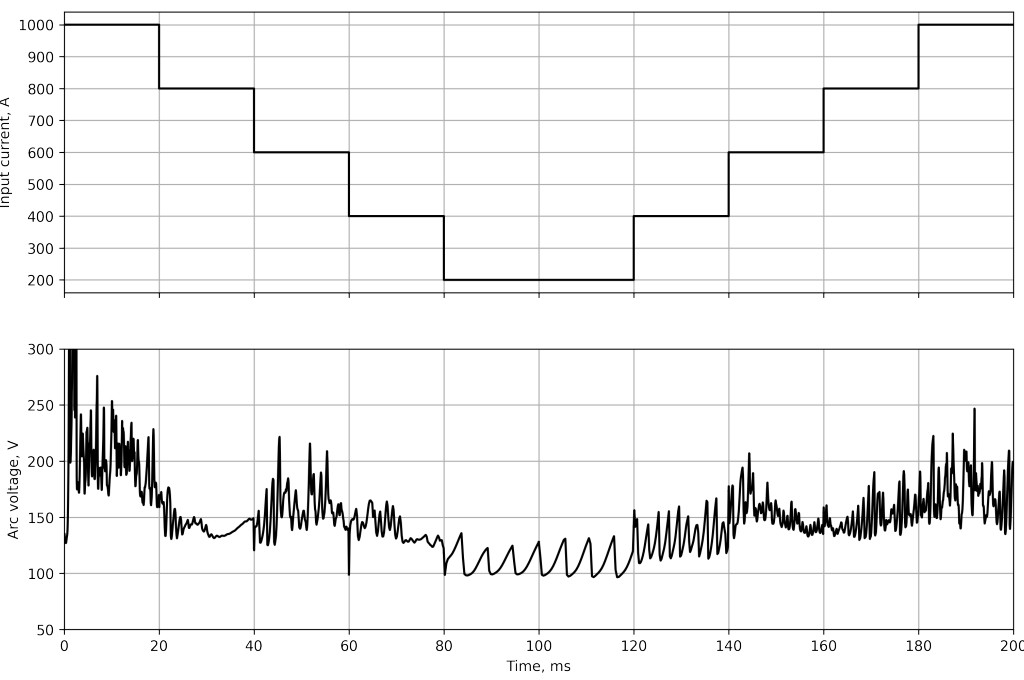

**Figure 8.** Arc voltages over the entire current sweep. $x_{CO}$ = 0.7, $L_a$ = 0.05 m.

In order to further quantify the results sections of the voltage time series, representative intervals of fixed current were sampled to obtain a representative average voltage and were also processed by Fourier transform. The power spectrum thus obtained was then analysed to find its peaks.

The peaks' magnitudes were taken to be indicative of the strength of the dynamics in the system, in particular if the arc was exhibiting steady or unsteady behaviour. An example of this is shown in Figure 9. Note that, since each current level is visited twice during a simulation, the pair of results from each level were averaged.

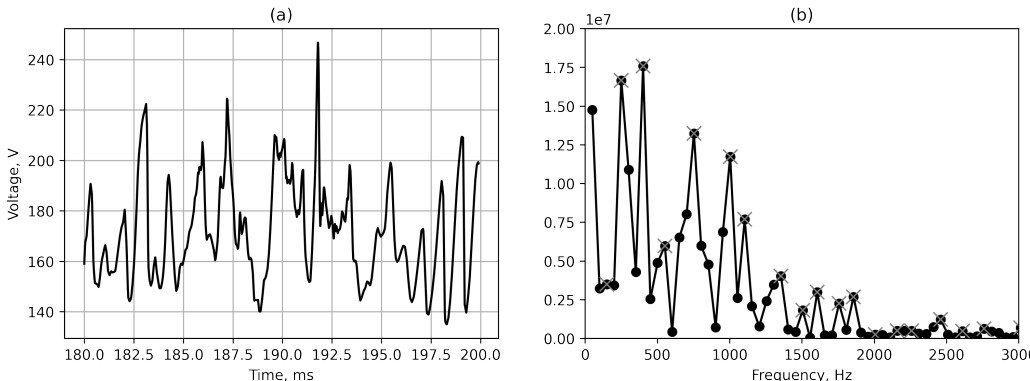

**Figure 9.** Arc voltages for a single current interval. $x_{CO} = 0.7$, $L_a = 0.05$ m and $I = 1000$ A: (**a**) Time series. (**b**) Spectral power with peaks identified by crosses.

Selected visualisations of the temperature and concentration of various plasma species are shown in Figures 10 and 11.

It is interesting to observe that, in this case, the arc has not chosen the shortest path between anode and cathode (vertically downward from the electrode tip) but is, instead, offset to one side. The strong turbulent jet creates very asymmetric flow and heat transfer patterns in the furnace vessel—at least over the short time periods modelled here.

The distribution of plasma species in the furnace vessel shows that, due to the high operating temperatures of the arc and the low dissociation energy of $H_2$, the bulk of the hydrogen is, in fact, present as individual atoms. CO only dissociates in the core of the arc jet, producing significant quantities of free C and O atoms in the same region. The most easily formed ion ($C^+$) is only present at levels of a few percent even in the hottest parts of the arc, suggesting that the plasmas in syngas arcs are not heavily ionised.

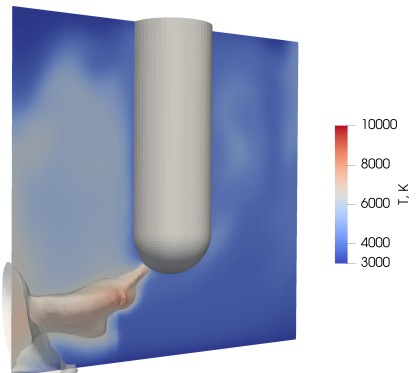

**Figure 10.** Temperature field at 200 ms with contour showing 6500 K isotherm. $x_{CO} = 0.7$, $L_a = 0.05$ m and $I = 1000$ A.

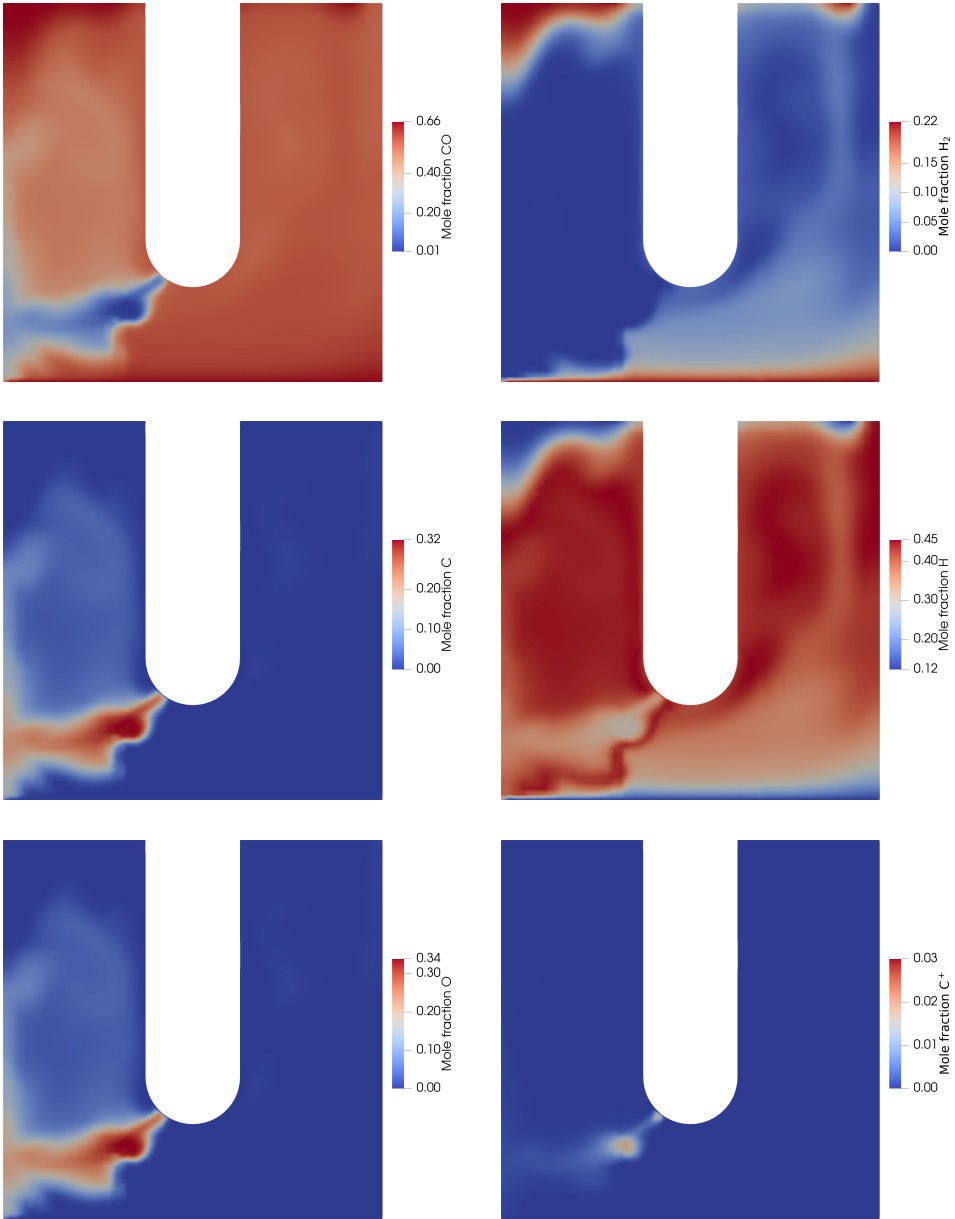

**Figure 11.** Plasma species concentration fields at 200 ms. $x_{CO} = 0.7$, $L_a = 0.05$ m and $I = 1000$ A.

The average voltages calculated from the MHD-CFD results at each $x_{CO}$, $L_a$ and $I$ value were then converted into furnace power by multiplying with the current to assess the operating regimes in which the model furnace is practically operable. The results are shown for all simulations in Figure 12. The axes of each graph refer to the operational variables of the furnace, and a separate graph is drawn for each combination of syngas chemistry.

Using a power limit of 70 kW, it can be seen that arcs in syngas mixtures will generally be limited to somewhat shorter arc lengths and/or lower currents when compared to the pure CO case ($x_{CO} = 1.0$); however, the effect is not particularly strong and is unlikely to prohibit operation entirely. A furnace that is operable with a CO atmosphere should, therefore, be operable with syngas mixtures, at least from a power-input perspective.

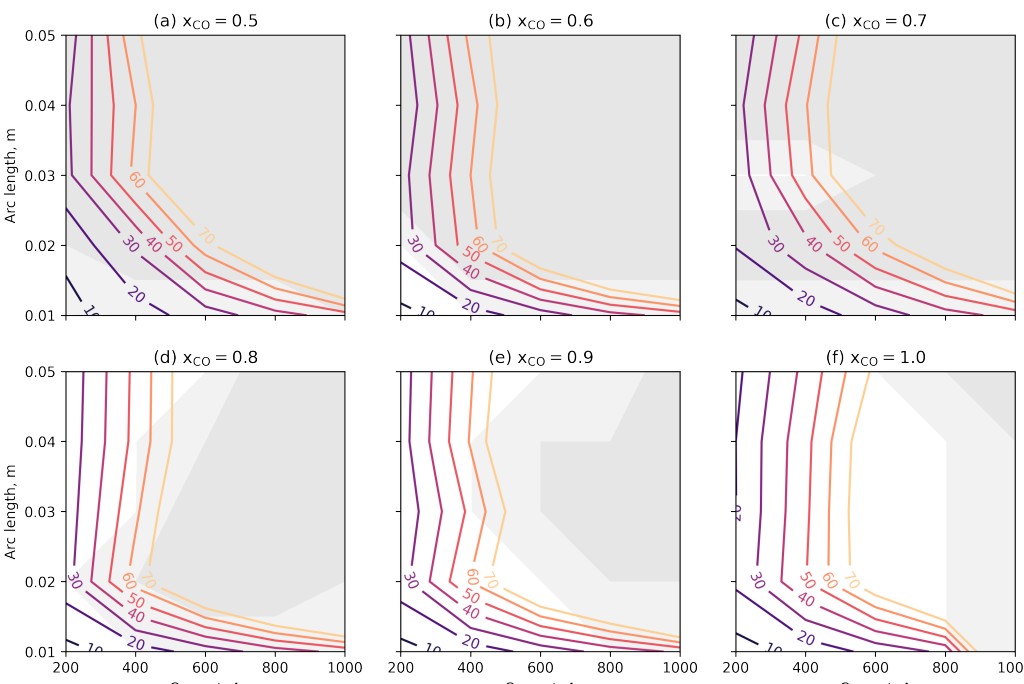

**Figure 12.** Furnace operability analysis, showing the power in kW as coloured contours and arc behaviour as shades of grey (white is steady state, and dark grey is fully transient).

In addition to the power input, the stability of the arc was examined using spectral analysis for each condition as described earlier. The results were then categorised in terms of "steady state" (no appreciable peaks in spectrum), "fully transient" (peaks observed during both the falling and rising current stages of each simulation) and "partially transient" (peaks observed during only one of the stages).

Transient behaviour can result in spikes of high arc voltage and may cause arc extinction in an operating furnace, thereby limiting its practical use. Figure 12 shows that the steady state window, which covers most of the parameter space for pure CO, retreats consistently as increasing amounts of $H_2$ are introduced into the system. For $x_{CO} \leq 0.7$, almost none of the simulations show steady state behaviour at any combination of current and arc length. This agrees with earlier observations on the plasma properties, which suggested that the additional non-linearities from the hydrogen reactions could lead to more instabilities in plasma arcs.

## 4. Conclusions

A modelling workflow for studying DC plasma arcs operating in syngas mixtures was successfully developed and demonstrated using simple test cases. The calculation of the plasma properties showed that syngas mixtures become increasingly non-linearly dependent on temperature as the fraction of hydrogen increases, in particular the thermal properties, such as $C_P$ and $\kappa$. This is anticipated to create an additional source of instability in the tightly coupled MHD system of plasma arcs.

A computational MHD-CFD model was then applied to the problem of arc behaviour in a small pilot-scale DC plasma arc furnace. Preliminary numerical testing showed that the model was in reasonable agreement with the empirical descriptions of plasma arcs from the literature. The parameter space of the problem was explored using simulations at a wide range of arc currents, arc lengths and syngas compositions. The results showed that, while increasing the fraction of hydrogen did not have a large impact on the electrical power delivered by the furnace, it did cause the arc behaviour to become more transient and unstable.

This effect potentially limits the operability window of DC furnaces used for syngas generation as such furnaces may be restricted to undesirable short-arc, low-current, high-voltage operation for a given power level; this can potentially make furnace control more sensitive and difficult.

Naturally, much work still remains to be performed in this area. In particular, validation of the model remains at an early stage and should be revisited once additional measured data from pilot experiments are available. In particular, accurate measurements of the relationship between voltage, current and arc length should be taken at various syngas compositions and compared with the model predictions in terms of both the dynamic behaviour and quantitative values. The design and implementation of more physically-realistic boundary conditions, especially at the anode and cathode conducting surfaces, would be of value in improving the accuracy of the model and permitting true mesh independence.

A deeper exploration of the plasma arc parameter space with computational simulations in order to develop reduced-order pragmatic or data-driven models of arcs in syngas mixtures will also be of interest. Ultimately, it is hoped that a combination of tools, such as those presented here, can provide practical and accessible engineering guidance in the design, operation and optimisation of arc furnaces and other units using similar technology in the future.

**Author Contributions:** Conceptualization, Q.G.R., T.P.K. and B.S.X.; methodology, Q.G.R.; software, Q.G.R.; writing—original draft preparation, Q.G.R.; writing—review and editing, T.P.K. and B.S.X.; project administration, Q.G.R., T.P.K. and B.S.X.; funding acquisition, Q.G.R., T.P.K. and B.S.X. All authors have read and agreed to the published version of the manuscript.

**Funding:** This research received no external funding.

**Acknowledgments:** This paper is published by permission of Mintek. The authors acknowledge the Centre for High Performance Computing (CHPC), South Africa, for providing computational resources to this research project.

**Conflicts of Interest:** The authors declare no conflict of interest.

## Abbreviations

The following abbreviations are used in this manuscript:

| | |
|---|---|
| DC | Direct current |
| LTE | Local thermodynamic equilibrium |
| MHD | Magnetohydrodynamic |
| CFD | Computational fluid dynamics |

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
