# Peer review of "A Computational Magnetohydrodynamic Modelling Study on Plasma Arc Behaviour in Gasification Applications"

_mca, doi:10.3390/mca28020060_

Round 1

Reviewer 1 Report

1) The physical model of syngas plasma should be simply displayed, such as, ionization reaction dynamics.

2) The mesh independence is not really independent.

3) Fig. 10 , it is difficult to read. Its ordinate should be the section position.

Author Response

Dear reviewer,

Many thanks for your review, we appreciate the time and effort taken. We have revised the paper as follows:

1) This has been addressed by including some simple examples of plasma reactions in the introduction section.

2) As the system is undergoing chaotic transient behaviour at the settings used, it is correct that it is harder to identify true mesh independence. We have softened our statement to "relatively mesh insensitive" rather than "mesh independent" in this section.

3) The choice of coordinate system used for display is intentional, since the arc length and arc current are furnace operation variables whereas the CO fraction is related to process chemistry. We have made a comment to this effect in the text describing Figure 10 (11 in the revised draft).

Best regards,
QG Reynolds

Reviewer 2 Report

This is a well-written paper simulating an arc with MHD. The title is appropriate, the abstract is clear, the use of figures is effective, the conclusions are supported by the contents, and the reference list is appropriate. I have only a few minor queries to the authors before giving a final recommendation.

1. Briefly justify the LTE is ok. In many arcs, the electrons may be much warmer than the ions.

2. How are Q_m and Q_r calculated?

3. Are you solving a species tranport equation for each species in line 145? If so, I should be given.

4. In line 187, maybe you mean that the mesh was "successively refined" instead of aggressively?

Author Response

Dear reviewer,

Many thanks for your review, we appreciate the time and effort taken. We have revised the paper as follows:

1) As these arcs are operating at moderate size and scale, and at atmospheric pressure, it is expected that they are predominantly in the thermal plasma regime and LTE is justified (except for the boundary sheath regions close to the electrode and anode surfaces, and possibly the cold outlying regions). The text has been amended to mention this.

2) A brief description of Q_m and Q_r has been added.

3) No, we do not solve individual species transport equations in the MHD model. LTE is assumed at all points in the mesh, and the transport and reaction kinetics of the individual species is assumed to be fast compared to the time scales being studied. The text has been amended to mention this.

4) Thank you for noticing this. The text has been revised.

Best regards,
QG Reynolds

Reviewer 3 Report

The manuscript need major revision

1.      In the Introduction, the authors need to describe not only what was done in previous works, i.e. it is also necessary to stress the main  results of each of them, otherwise the reader will not properly understand how the present work compares to the state-of-the-art on the subject. Once the state-of-the-art is characterized, the authors should state the objectives of their work based on what lacks to be done in the subject, emphasizing its novelty and possible answers to open questions. All references should be cited in uniform order. https://doi.org/10.1039/d2ra08197khttps://doi.org/10.1016/j.csite.2021.100870

2.     Some parameters are defined incorrectly

3.     There are concerns about the grammar, usage, and overall readability of the manuscript. It is not possible to mention all the grammatical errors here. The authors are recommended to carefully read the manuscript again and remove all mistakes

4.     The abstract should be revised, it should be presented in a way to reflect your research work, connect the title to the abstract and abstract to introduction

5.     Give a broader view of the literature on the topic and the current state-of-the-art;

6.     Tables and basic equations are presented without details and references

7.     clarify and discuss the novelty and the significance of the results obtained here, and compare them with those available in the literature, also including discussions on potential applications;

8.     compare the results with previous work

9.     What is the physical meaning of boundary conditions, why such transformations are used ?

10.  Include detail of applied scheme, why such method used

11.  There are various mistakes in writing and defining the symbols used in the manuscript. Please rectify these mistakes.

12.  Include  nomenclature section including SI units

13.  The results and discussion section must be improved. Physical justification behind the results must be provided for each and every graph. Authors have mentioned only the increasing/decreasing trend of the curves. Validated your results with previous methods/results

14.  Conclusion section need improvement  

Author Response

Dear reviewer,

Many thanks for your review, we appreciate the time and effort taken. In light of your extensive comments we resubmitted our paper for internal and external review by native English-language speakers with expertise in the field. I have summarised and collated the responses below.

1)The introduction has been revised with some additional background and description of the waste coal gasification process, contextualising the problem. The reference ordering is automated in the MDPI LaTeX template and is out of the authors' control - we will take it up with the editors during the publication process.

2) The parameter definitions were reviewed and apart from minor additions and clarifications, no obvious errors were found.

3) Langauge review by expert native-English speakers found no major issues with the grammar. Some instances of clumsy or unclear language were identified and corrected in the revised manuscript.

4) Minor revisions were made to the abstract to improve the storyline.

5) The literature on the subject of DC arc furnaces for waste coal gasification is extremely limited, as this is a new concept. It was felt by all other reviewers that the background cited is already sufficient to give the necessary context.

6) All relevant equations are referenced with citations to prior work. It was not felt to be necessary to provide supporting citations for fundamental governing equations such as the Navier-Stokes or energy conservation equations - these are well-established in the field, and readers of this paper should be familiar with them already.

7) The industrial application of this work has been further contextualised with additional material in the introduction section. As mentioned on several occasions in the paper, literature results for this particular application are currently limited due to its novelty - we have added some commentary in the conclusions which expands on the future experimental work to be done to address this situation.

8) Please see 7). We have compared the model to available literature via the work of Bowman on general DC arc behaviour, however, sufficient data specific to DC plasma arc furnaces operating with syngas atmospheres does not exist at this time.

9) Additional text has been included at Table 2 which describes the physical interpretation of the boundary conditions.

10) The numerical scheme applied to the solution of the MHD equations has been described in detail in previous work - it is cited in this paper, and available open access. It was felt to be too close to plagiarism to repeat this information again.

11) No issues with mathematical symbols were identified by the other reviewers. This may be an issue with with the rendering of your particular copy from MDPI? We will make sure to check this carefully prior to publication.

12) After consultation with our re-review panel, they have advised that adding a nomenclature would make the paper unnecessarily verbose. We have however revised some of the symbol definitions in the text where they first appear to make sure they are clear.

13) Please see 7) and 8). Reliable experimental results do not currently exist for this application, which is why the focus is on broad, qualitative system behaviour and not specific, quantitative engineering detail. Future papers on this subject will of course develop toward this goal, but the state of the art is not there yet.

14) The conclusion section has been revised and some changes to the text have been made to improve clarity.

Best regards,
The authors

Reviewer 4 Report

The model of the computational magnetohydrodynamic solver for electromagnetically coupled flows and the behavior of three-dimensional transient simulations of arc DC discharge formation and dynamics in response to different plasma gas compositions and external conditions is studied. The work is relevant for the development of the design, operation, and optimization of arc furnaces. Model tested on the 100 kVA DC plasma arc furnace pilot facility at Mintek. Preliminary numerical testing showed that the model was in reasonable agreement with empirical descriptions of plasma arcs from literature.
But there are several remarks:
1. The declared model implies magnetohydrodynamic interaction, the system of equations contains the magnetic induction vector B, the Lorentz force j x B, the induced current term u x B, but nowhere is it explained what kind of field it is, what magnitude, how it arises, it is external or induced by gas-discharge current?
2. It is not clear in which direction the Lorentz force acts in this case, because it is a vector quantity?
3. Would it be nice to show in a separate figure how both the magnetic field and the Lorentz force are directed? And also to explain the mechanism of influence of MHD interaction on the process of operation of an electric arc furnace.

Author Response

Dear reviewer,

Many thanks for your review, we appreciate the time and effort taken. We have revised the paper as follows:

1) The magnetic field B is not external, it is self-induced by the electric current per the relationships in equation (8). The text in this section has been amended to make this clearer.

2) and 3) Thank you for this good suggestion. A schematic figure has been added to show the origin and direction of the Lorentz force in the arc.

Best regards,
QG Reynolds